# Effects of Radix Polygalae on Cognitive Decline and Depression in Estradiol Depletion Mouse Model of Menopause

**Gaeul Han [1,†], Junhyuk Choi [1,†], Seung-Yun Cha [1], Byung Il Kim [1] , Hee Kyung Kho [1], Maeng-Jin Jang [1], Mi Ae Kim [1], Sungho Maeng [1,2,*] and Heeok Hong [3,*]**

1. Graduate School of East-West Medical Science, Kyung Hee University, Yongin-si 17104, Korea; hkm1356@naver.com (G.H.); ketcpn6914@naver.com (J.C.); suguk93@khu.ac.kr (S.-Y.C.); sleep-dr@hanmail.net (B.I.K.); rhgmlrud@khu.ac.kr (H.K.K.); supermaeng21@khu.ac.kr (M.-J.J.); zoe1001@naver.com (M.A.K.)
2. Department of Gerontology (AgeTech-Service Convergence Major), Graduate School of East-West Medical Science, Kyung Hee University, Yongin-si 17104, Korea
3. Department of Animal Science and Technology, Konkuk University, Seoul 05029, Korea
* Correspondence: jethrot@khu.ac.kr (S.M.); hhong58@konkuk.ac.kr (H.H.); Tel.: +82-31-201-2916 (S.M.); +82-2-2049-6274 (H.H.)
† These Authors contributed equally.

**Abstract:** Postmenopausal syndrome refers to symptoms caused by the gradual decrease in female hormones after mid-40 years. As a target organ of estrogen, decrease in estrogen causes various changes in brain function such as a decrease in choline acetyltransferase and brain-derived neurotrophic factor; thus, postmenopausal women experience cognitive decline and more depressive symptoms than age-matched men. Radix Polygalae has been used for memory boosting and as a mood stabilizer and its components have shown neuroprotective, antidepressant, and stress relief properties. In a mouse model of estrogen depletion induced by 4-vinylcyclohexene diepoxide, Radix Polygalae was orally administered for 3 weeks. In these animals, cognitive and depression-related behaviors and molecular changes related to these behaviors were measured in the prefrontal cortex and hippocampus. Radix Polygalae improved working memory and contextual memory and despair-related behaviors in 4-vinylcyclohexene diepoxide-treated mice without increasing serum estradiol levels in this model. In relation to these behaviors, choline acetyltransferase and brain-derived neurotrophic factor in the prefrontal cortex and hippocampus and bcl-2-associated athanogene expression increased in the hippocampus. These results implicate the possible benefit of Radix Polygalae in use as a supplement of estrogen to prevent conditions such as postmenopausal depression and cognitive decline.

**Keywords:** estrogen; radix polygalae; cognition; depression; VCD

## 1. Introduction

Menopause is an interruption of the menstrual cycle by the depletion of ovarian follicles, which causes a gradual depletion of circulating estrogen [1]. Estrogen deficiency can cause postmenopausal symptoms such as facial flushing, osteoporosis, and genitourinary atrophy [2,3]. Estrogen is an important steroid hormone involved in the functional regulation of reproductive organs but also regulates many organ systems including the brain [4,5]. It affects cholinergic and serotonergic neurotransmitter systems and provides neuroprotection against excitotoxicity and β-amyloid [5]. As a result, normal brain functions rapidly deteriorate in the postmenopausal state of estrogen deficiency [3]. Many studies have demonstrated that estrogen supplementation in the postmenopausal brain can provide protection from emotional and neurodegenerative disorders [6].

Women have a higher risk of developing Alzheimer's disease (AD) than men, perhaps because of the abrupt decrease in estrogen levels in the postmenopausal period [7]. One

feature of AD is the reduction of choline acetyltransferase (ChAT), an enzyme that synthesizes acetylcholine [8]. Estrogen regulates cholinergic neurotransmission in the brain and promotes choline uptake and ChAT in the frontal lobe and hippocampus [9]. The main source of cholinergic projections to the prefrontal cortex and hippocampus origins from the medial septal nucleus (MS), diagonal band of Broca (DBB), nucleus basalis magnocellularis (NBM) in the ventral pallidum in association of learning, memory, and attentional processes [10]. Reduced cholinergic functions in these regions are associated with age-related cognitive decline and dementia [11]. Estrogen is related to the cholinergic functions, such as ovariectomized animals that have shown reduced choline uptake, ChAT activity, and ChAT mRNA expression [9]. Estrogen supplement delayed brain aging and reduced the risk of AD in postmenopausal women [8]. Estradiol treatment also increased ChAT activity in MS, DBB, NBM regions [12]. However, currently available AD medicine such as donepezil increases the amount of acetylcholine in the brain and improves cognitive functions in AD patients, however, it does not prevent the progression of AD [13]. Moreover, estrogen supplement failed to prevent dementia progression, especially after menopause [14,15]. Therefore, an improvement in the method to prevent cognitive decline in menopause is required.

The incidence of depression is higher in physiologically unstable endocrine periods in life such as puberty, pregnancy, postpartal, and the premenstrual phase [16]. In addition, the unstable and irregular patterns of hormone production during perimenopause increase women's vulnerability to mood disorders [17]. Low estrogen levels contribute to reduced vigilance, which in turn is correlated with depressive and menopausal symptomatology of women [18,19]. Estrogen has antidepressant effects in depression animal models [20]. It decreases the susceptibility for depression and improves the antidepressant effect of selective serotonin reuptake inhibitors (SSRI) [21,22]. Additionally, estrogen increases the expression of brain-derived neurotrophic factor(BDNF), which is related to emotional stability [20,23]. However, there are contradictory reports of the positive effect of estrogens on mood disorders [24]. Furthermore, it is argued that estrogen supplement can increase the risk of thromboembolism and cancer [15]. Meanwhile, postmenopausal depression is more severe, has a more insidious course, and is more resistant to conventional antidepressants in comparison with premenopausal depression in women [25].

The hypothalamus–pituitary–adrenal gland (HPA) axis regulates the body's response to stress by stimulating the release of glucocorticoid hormones [26]. The glucocorticoid receptor (GR) of the hippocampus provides negative feedback control of the HPA axis in response to circulating glucocorticoids [27]. Research suggests that fluctuations in ovarian hormones and derived neurosteroids result in HPA axis dysfunction, thereby increasing vulnerability to depression [28]. A hyperactive HPA axis is a common feature in depression [29]. Reduced GR activity in the hippocampus can cause a hyperactive HPA axis and downregulate BDNF expression in brain circuits [30,31]. As ovariectomy increases GR sensitivity, emotional liability during the perimenopausal period may also be associated with GR hypersensitivity [32]. Meanwhile, Bcl-2-associated athanogene (BAG1) is a co-chaperone of the GR assembly line that reduces the activity of GR and cortisol–GR complex-induced gene transcription [2]. In the literature, BAG1 assisted recovery from stressful adverse effects and from behavioral disorders such as mania and depression in mice [33]. So, reagents that increase BAG1 may aid with diseases caused by GR hypersensitivity.

Radix Polygalae (RP) is the root of *Polygala tenuifolia* Willdenow and has traditionally been used as an expectorant, tranquilizer, and memory booster [34–36]. Recent studies have demonstrated that it may have effects on dementia prevention, neuroprotection, and antidepressant [37]. RP improved memory and increased ChAT activity in β-amyloid induced learning and memory-impaired animal models [38]. RP also increased BDNF levels and CREB phosphorylation via the CaMKII and ERK1/2 pathway [20]. Tenuigenin, which is an active ingredient of RP, protected hippocampal neurons against neurotoxicity [39]. Another ingredient, polygalasaponin XXXII, also improved hippocampal-dependent learning and

memory [40]. According to depression, RP and its component DISS (3,6′-disinapoyl sucrose) showed antidepressant-like effects in chronically mild stressed (CMS) animals [34,41]. Moreover, there is evidence that RP has a rapid-onset antidepressant action [35]. Stress-related effects were also demonstrated such as resistance to social isolation and restraint and hippocampal BAG1 induction after RP administration [2,42]. As having memory-improving, antidepressant, and mood-stabilizing effects, RP is expected to be an effective remedy against menopausal symptoms.

Menopause is the depletion of ovarian primitive follicles, eventually leading to ovarian aging [43]. Ovariectomy (OVX), a widely used method in postmenopausal studies in animals, causes rapid disruption of ovarian function but is physiologically different than menopause because other hormones such as androgens are also devoid in this model [44]. In contrast, 4-vinylcyclohexene diepoxide (VCD)-induced estradiol depletion is more physiologically like menopause. VCD selectively destroys ovarian primitive follicles in mice and the secretion of ovarian hormones does not abruptly stop, and androgens are produced in residual ovarian tissues [45].

This study hypothesized that RP may improve cognitive decline and depression caused by estrogen deficiency. Therefore, VCD was administered for 20 days on 4 weeks old mice to establish a chemically induced estradiol depletion mouse model. RP was fed orally for 3 weeks and its effects were evaluated by behavioral and biochemical measurements.

## 2. Materials and Methods

### 2.1. Reagents

Radix Polygalae was purchased from Kyung Hee Hanyak (Seoul, Korea) and authenticated by Professor Jae-Hwan Lew. A voucher specimen was deposited in the Kyung Hee University College of Oriental Medicine Herbarium (KHUS-r002h). To obtain extracts of RP, the first extraction was performed by heating 1.5 kg of RP in 70% ethanol at 80 °C for 1 h. The remains of RP were re-extracted for 40 min under the same condition. Extracts were filtered and concentrated under reduced pressure and spray dried. The yield of the extract was 28% (*w/w*). HPLC fingerprint of the extract showed a stable amount of norharmane, DISS, TMCA (3,4,5-trimethoxycinnamic acid), and tenuifolin as active ingredients [46]. For experimental use, the powdered extract was dissolved in distilled water.

### 2.2. Animals

Four weeks old female C57BL/6 mice (Nara Biotech, Seoul, Korea) were used in the experiment. The experimental procedures were performed in accordance with the animal care guidelines of Kyung Hee University in Institutional Animal Care and Use Committee (KHUASP(SE)-19-038). Animals were housed under a controlled environment: temperature ($23 \pm 2$ °C), humidity ($55 \pm 10$%), and lighting (07:00–19:00 h) conditions, with food and water made available ad libitum. Mice were randomly divided into six groups: Control ($n = 8$): Vehicle (sesame oil) i.p.; VCD ($n = 6$): VCD 160 mg/kg i.p.; VCD + E2 ($n = 7$): VCD 160 mg/kg i.p + estradiol 100 μg/kg i.p.; VCD + RP1 ($n = 7$): VCD 160 mg/kg i.p + RP 1 mg/kg p.o.; VCD + RP10 ($n = 7$): VCD 160 mg/kg i.p + RP 10 mg/kg p.o.; and VCD + RP100 ($n = 7$): VCD 160 mg/kg i.p + RP 100 mg/kg p.o.

### 2.3. VCD, E2 and RP Treatment

Mice were intraperitoneally (i.p.) injected with VCD (94956, Sigma, St. Louis, MO, USA) for 20 consecutive days (160 mg/kg). Sesame oil (vehicle solution) (sc-215848, Santa Cruz, Dallas, TX, USA) was injected into controls. VCD 160 mg/kg for 20 days was determined to accelerate estradiol depletion by the toxicity on primordial follicles [47]. At 4 weeks of age, toxicity is limited to the ovary only, but in older age (12 weeks) this dose induced toxicity to other organs [45]. E2 and RP treatment was initiated 5 weeks after the termination of VCD treatment (perimenopausal period). E2 was injected i.p. and RP was fed by oral gavage both daily for 3 weeks. Mice in the control and VCD groups were

fed distilled water by oral gavage for the same period. After the treatment, cognitive and depression-like behaviors were measured. Figure 1 depicts the experimental outline.

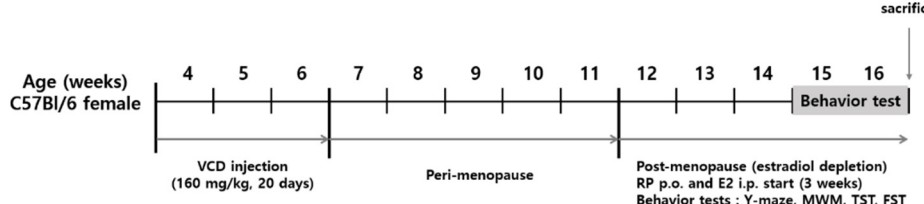

**Figure 1.** Schedule of VCD, E2, RP treatment, behavior tests, and tissue collection.

### 2.4. Vaginal Cytology

To confirm VCD-induced estradiol depletion, vaginal cells were collected 10 days before the initiation of RP treatment (10th week), daily at 10 a.m. to observe the estrous cycle. To collect vaginal cells, the vaginal canal was flushed with saline by pipetting. During the flushing, care was taken not to enter the pipette tip into the vaginal canal. The fluid was spread on a slide and was allowed to air dry. Then the slide was immersed in 0.1% crystal violet solution for 1 min, followed by washing in ddH$_2$O twice for 1 min each. After the slides were dried sufficiently, slides were coverglassed and observed under an optical microscope. Mice that received VCD but maintained their normal estrous cycles for more than 10 days were excluded from the experiment. Initially, there were 8 mice per group, and 2 from the VCD group, and 1 each from the other groups failed to meet the criteria and were excluded.

### 2.5. Y-maze

Y-maze was performed to test working memory. Each arm was labeled—A, B, and C—and after putting the mouse in the center of the maze, the movement path of the mouse was recorded for 10 min. This test was based on the principle that mice prefer to explore novel places, which would result in them entering the three different paths one by one. The movement path of the mouse was recorded and analyzed using SMART 3.0 VIDEO TRACKING Software (Panlab, Barcelona, Spain). The *alternation triplet* (%) was calculated using the following equation:

$$Alternation\ triplet = \frac{Alternation}{Total\ entrie - 2} \times 100$$

### 2.6. Morris Water Maze

The Morris water maze (MWM) test was performed to measure spatial memory. A round metal tank 180 cm diameter and 45 cm high was filled with water (22 ± 2 °C, depth 30 cm). A platform was submerged 0.5 cm under the water and was hidden to the animals by making the water nontransparent by adding white-colored paint. Four symbolic figures were attached to the wall in well-sighted places to the mice. The training lasted 5 days and was performed four times per day. During the 4 daily trials, the starting point differed on each trial. Training sessions continued until the mouse climbed up to find the platform, and after 1 min, if the platform was not found, the mouse was guided to the platform. The time to reach the platform was measured. On the 6th day, the probe test was performed without the platform. The trajectories of the swimming of the mice were tracked for a total of 90 s and the time spent in the quadrant where the platform was placed was measured. For measurement and analysis, SMART 3.0 was used.

### 2.7. Forced Swim Test

The forced swim test (FST) is a despair-related behavioral measurement to screen antidepressant-like effects. A cylindrical test chamber was filled with tap water to more than half of the height of the chamber. Mice were placed in the chamber for 6 min, and the immobile time during the last 4 min measured by two researchers. Immobility was

considered when there were no movements except movements related to breathing or hind limbs movements to maintain floating.

### 2.8. Tail Suspension Test

The tail suspension test (TST) is a measurement of despair behavior widely applied to screen antidepressant-like effects. Mice were suspended by the tail using tape at a height of 40 cm. The movements were videotaped, and the immobility time was measured for 6 min by two researchers.

### 2.9. Measurement of Serum Estradiol Concentration

Under deep anesthesia, blood was obtained from the inferior vena cava. Coagulated blood was centrifuged at $13,000 \times g$ for 10 min to separate serum. Estradiol concentration was determined with the 17-beta Estradiol ELISA Kit (ab108667, Abcam, Cambridge, UK) following the manufacturer's manual. The optical density of the samples was measured by VICTOR X3 (Perkin-Elmer Korea, Seoul, Korea) microplate reader.

### 2.10. Western Blotting

After blood sampling, mice were decapitated and the prefrontal cortex and hippocampus were separated and stored at $-80\ ^{\circ}C$ for later use. Tissues were homogenized with lysis butter containing 1% each of Phosphatase Inhibitor Cocktail 2, 3 (P5726, P044, Sigma-Aldrich, St. Louis, MO, USA) in PRO-PREP™ (17081, iNtRON Biotechnology, Seongnam, Korea) by a homogenizer and centrifuged at $16,000 \times g$ at $4\ ^{\circ}C$ for 10 min. Protein concentration was measured using DC™ Protein Assay Reagent A, B, S (5000113, 5000114, 5000115, BIO-RAD, Hercules, CA, USA).

After electrophoresis using 10% or 12% SDS-polyacrylamide gel, the gel phase proteins were transferred to a PVDF membrane (162-0177, BIO-RAD). Membranes were incubated for 1 h at room temperature in 5% nonfat dry milk (170-6404, BIO-RAD) in TBS-T (0.1% tween 20 in TBS) for blocking. Primary antibodies were diluted in 5% nonfat dry milk in TBS-T and incubated at $4\ ^{\circ}C$ overnight. The next day, membranes were washed 3 times for 10 min with TBS-T. Secondary antibodies were diluted with 5% nonfat dry milk in TBS-T and incubated for 1 h at room temperature. After washing three times for 10 min with TBS-T, luminescence was induced using Pierce™ ECL Western Blotting Substrate Kit (32106, Thermo Fisher Scientific, Waltham, MA, USA) and photographed with EZ-Capture MG (ATTO, Tokyo, Japan). Band density was analyzed using the ImageJ software (NIH, Bethesda, MD, USA).

Antibodies used in this experiment were as follows: ChAT (PA5-79038, 1.1000, Invitrogen, Carlsbad, Germany), BDNF (OSB00017W, 1:1000 Invitrogen), BAG1 (sc-8348, 1:1000, Santa Cruz), GR (ab3578, 1:1000, Abcam), Beta-actin (sc-47778, 1:5000, Santa Cruz), Anti-Rabbit IgG (H + L) HRP conjugate (W4011, 1:1000, Promega, Madison, WI, USA), and Anti-Mouse IgG (H + L) HRP conjugate (W4021, 1:5000, Promega).

### 2.11. Statistical Analysis

Experiment data were expressed as the mean $\pm$ SEM. ANOVA and repeated measures ANOVA followed by Tukey's HSD test by using SPSS 25 (SPSS Inc., Chicago, IL, USA) were used for statistical analysis. Significance was set at $p < 0.05$.

## 3. Results

### 3.1. Disruption of the Estrous Cycle in VCD-Induced Estradiol Depletion Mice

The estrus cycle is a periodic change in proestrus, estrus, metestrus, and diestrus phases. Each stage is distinguished by the type and proportion of cells observed in the vaginal smear. Proestrus is characterized by relatively uniform-shaped small, nucleated cells. During the estrus, most cells become keratinized epithelial cells. Neutrophils appear and keratinized epithelial cells reduce at metestrus, and the number of neutrophils and epithelial cells reduces at diestrus compared with metestrus. After 3 weeks of VCD

treatment and 5 weeks of the perimenopausal period (indicated in Figure 1), vehicle-treated mice remained to show the regular cycling pattern of the estrus cycle (Figure 2A), but lack of cycling patterns of estrus stages in the VCD-treated groups of mice (Figure 2B). These findings indicated an arrest of estrus cycle after VCD treatment.

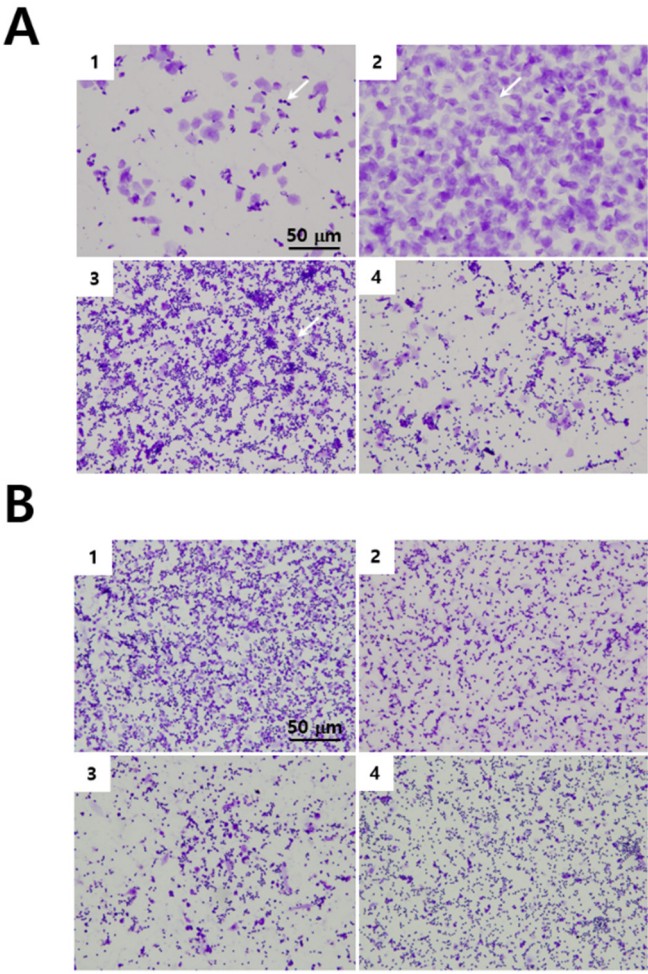

**Figure 2.** Vaginal cytology of normal and VCD-treated mice. Vaginal smear stained with crystal violet. (**A**) Vehicle-treated mice completed all 4 estrous stages (proestrus: A-**1**, estrus: A-**2**, metestrus: A-**3**, diestrus: A-**4**). A-**1** (arrows): small nucleus epithelial cells, A-**2** (arrows): keratinized epithelial cells, A-**3** (arrows): surrounded by keratinized epithelial cells. (**B**) VCD-treated mice showed the characteristics of diestrus and meta-to-diestrus stages by the presence of small nuclear epithelial cells and neutrophils.

### 3.2. RP Had No Effect on the Uterus Weight and Serum Estradiol Concentration in VCD-Treated Mice

Serum estradiol concentration was measured 13 weeks after the initiation of VCD treatment (Figure 3A; $n$ = 5~6/group). Estradiol concentration showed group difference [$F(5,26)$ = 10.5, $p < 0.001$]. Compared with the control group, estradiol concentration decreased in VCD ($p$ = 0.001), VCD + RP1 ($p$ = 0.003), VCD + RP10 ($p$ = 0.005), and VCD + RP100 ($p < 0.001$), but not in VCD + E2 ($p$ = 0.22). Compared with the VCD group, estradiol concentration increased in VCD + E2 ($p < 0.001$) but not in the RP-treated groups.

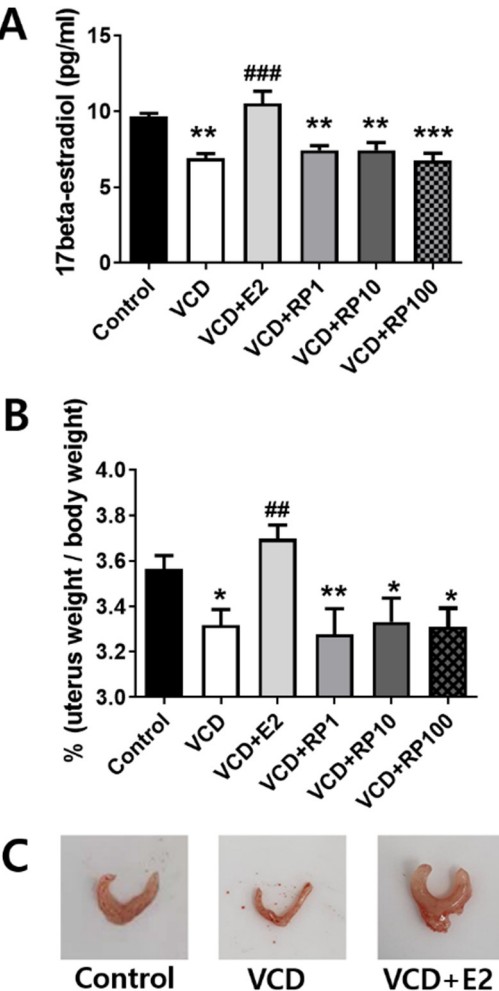

**Figure 3.** Effect of E2 and RP on uterus weight and serum estradiol concentration in VCD-treated mice. (**A**) Serum concentration of estradiol (pg/mL). Compared with the control group, estradiol concentration decreased in VCD, VCD + RP1, VCD + RP10, and VCD + RP100, but not in VCD + E2. Compared with the VCD group, estradiol concentration increased in VCD + E2 but not in VCD + RP treated three groups. (**B**) Uterus-to-body weight ratio (%). Compared with the control group, ratio decreased in the VCD group. Compared with the VCD group, the ratio increased in VCD + E2 but not in RP-treated groups. (**C**) Representative images of the uterus from Control, VCD, and VCD + E2. All data are the mean ± SEM. Control: Vehicle (sesame oil) i.p.; VCD: VCD 160 mg/kg i.p.; VCD + E2: VCD 160 mg/kg i.p + estradiol 100 μg/kg i.p.; VCD + RP1: VCD 160 mg/kg i.p + RP 1 mg/kg p.o.; VCD + RP10: VCD 160 mg/kg i.p + RP 10 mg/kg p.o.; and VCD + RP100: VCD 160 mg/kg i.p + RP 100 mg/kg p.o.; * $p < 0.05$, ** $p < 0.01$, *** $p < 0.001$ vs. Control, $^{##}$ $p < 0.01$, and $^{###}$ $p < 0.001$ vs. VCD. ANOVA, Tukey's HSD post hoc test.

As uterine atrophy weight was suggested in VCD-treated mice, uterus to body weight ratio was measured (Figure 3B, $n = 6$~$8$/group). The ratio showed group difference [$F(5,33) = 4.4$, $p = 0.004$]. Compared with the control group, weight ratio decreased in the VCD ($p = 0.025$), VCD + RP1 ($p = 0.007$), VCD + RP10 ($p = 0.028$), and VCD + RP100 ($p = 0.017$) groups, but not in VCD + E2 ($p = 0.422$). Compared with the VCD group, the weight ratio increased in VCD + E2 ($p = 0.006$) but not in the RP-treated groups (Figure 3B). These results showed that RP does not have an estrogenic hormonal effect.

### 3.3. RP Improved Working and Spatial Memories in VCD-Induced Menopausal Mice

Working and spatial memory was measured with the Y-maze and MWM, respectively (Figure 4; $n = 6$~$8$/group). The alternation triplet ratio in the Y-maze showed

group difference [F(5,36) = 3.95 $p$ = 0.011]. Compared with the control group, the working memory performance decreased in the VCD group ($p$ = 0.021).  Compared with the VCD group, the working memory performance increased in the VCD + E2 group ($p$ = 0.007), VCD + RP1 ($p$ = 0.042), and VCD + RP100 ($p$ = 0.023), and non-significant trend in VCD + RP10 ($p$ = 0.064) group (Figure 4A).

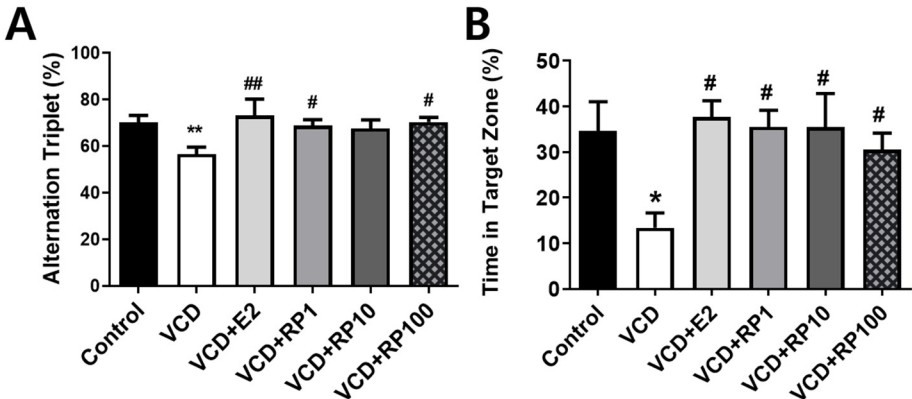

**Figure 4.** Effect of E2 and RP on working and spatial memory in VCD-treated mice. (**A**) Alternation triplet ratio (%) in the Y-maze test. Compared with the control group, the ratio decreased in VCD. Compared with the VCD group, the ratio increased in the VCD + E2, VCD + RP1, and VCD + RP100, and non-significantly in VCD + RP10 ($p$ = 0.064) group. (**B**) Target quadrant time (%) in the MWM probe test.  Compared with the VCD group, the %time increased in the VCD + E2, VCD + RP1, VCD + RP10, and VCD + RP100. Data are the mean ± SEM. Control: Vehicle (sesame oil) i.p.; VCD: VCD 160 mg/kg i.p.; VCD + E2: VCD 160 mg/kg i.p + estradiol 100 μg/kg i.p.; VCD + RP1: VCD 160 mg/kg i.p + RP 1 mg/kg p.o.; VCD + RP10:  VCD 160 mg/kg i.p + RP 10 mg/kg p.o.; and VCD + RP100: VCD 160 mg/kg i.p + RP 100 mg/kg p.o; * $p$ < 0.05, ** $p$ < 0.01 vs. Control, [#] $p$ < 0.05, and [##] $p$ < 0.01 vs. VCD. ANOVA, Tukey's HSD post hoc test.

The time spent in the target quadrant in the MWM showed group difference [F(5,36) = 4.1, $p$ = 0.033]. Compared with the control group, %time decreased in VCD group ($p$ = 0.013). Compared with the VCD group, the %time increased in the VCD + E2 group ($p$ = 0.022), VCD + RP 1 mg/kg ($p$ = 0.023), VCD + RP 10 mg/kg ($p$ = 0.011), and VCD + RP 100 mg/kg ($p$ = 0.049 (Figure 4B). These results show that estrogen and RP can improve VCD-induced defects of cognitive functions.

### 3.4. RP Improved Depression-Like Behaviors in VCD-Induced Menopausal Mice

Despair behaviors as a measurement of depression-like response was measured with the FST and TST (Figure 5; $n$ = 6~8/group). The time spent immobile in the FST showed group difference [F(5,36) = 10.8, $p$ < 0.001]. Compared with the control group, immobility increased in the VCD group ($p$ = 0.001) and VCD + RP1 ($p$ < 0.001). Compared with the VCD group, the immobility decreased in VCD + E2 ($p$ = 0.01), VCD + RP10 ($p$ < 0.001), and VCD + RP 100 ($p$ < 0.001) (Figure 5A).

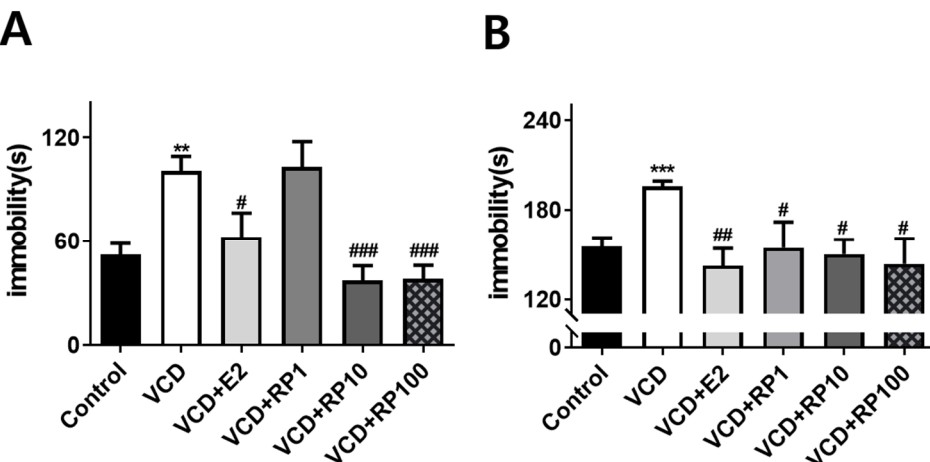

**Figure 5.** Effect of E2 and RP on despair behaviors in VCD-treated mice. (**A**) Immobility(sec) in the FST. Compared with the control group, immobility increased in the VCD group. Compared with the VCD group, the immobility decreased in VCD + E2, VCD + RP10, and VCD + RP100 group. (**B**) Immobility in the TST. Compared with the control group, immobility increased in the VCD group. Compared with the VCD group, the immobility decreased in VCD + E2, VCD + RP1, VCD + RP10, and VCD + RP100 group. All data are the mean ± SEM. Control: Control: Vehicle (sesame oil) i.p.; VCD: VCD 160 mg/kg i.p.; VCD + E2: VCD 160 mg/kg i.p + estradiol 100 μg/kg i.p.; VCD + RP1: VCD 160 mg/kg i.p + RP 1 mg/kg p.o.; VCD + RP10: VCD 160 mg/kg i.p + RP 10 mg/kg p.o.; and VCD + RP100: VCD 160 mg/kg i.p + RP 100 mg/kg p.o; ** $p < 0.01$, *** $p < 0.001$ vs. Control, # $p < 0.05$, ## $p < 0.01$, and ### $p < 0.001$ vs. VCD. ANOVA, Tukey's HSD post hoc test.

Immobility in the TST, also showed group difference [$F_{(5,36)}$ = 2.3, $p = 0.044$]. Compared with the control group, immobility increased in VCD ($p = 0.038$). Compared with the VCD group, the immobility decreased in VCD + E2 ($p = 0.002$), VCD + RP1 ($p = 0.015$), VCD + RP10 ($p = 0.035$), and VCD + RP100 ($p = 0.03$) (Figure 5B). These results show that estrogen and RP can improve VCD-induced depression-like behaviors.

*3.5. Effect of RP on the Prefrontal Cortex and Hippocampal Expression of ChAT and BDNF in VCD-Induced Menopausal Mice*

Key molecules of cognitive and emotional regulation were measured in the prefrontal cortex and hippocampus (Figure 6; $n$ = 6~8/group). ChAT expression in the prefrontal cortex showed group difference [$F_{(5,36)}$ = 2.6, $p = 0.039$]. Compared with the control group, the intensity ratio decreased in VCD group ($p = 0.021$). Compared with the VCD group, the intensity ratio increased in VCD + E2 ($p = 0.041$), VCD + RP1 ($p = 0.004$), VCD + RP10 ($p = 0.017$), and VCD + RP100 ($p = 0.003$) (Figure 6B). Additionally, ChAT expression in the hippocampus showed group difference [$F_{(5,36)}$ = 4.2, $p = 0.006$]. Compared with the control group, the intensity ratio decreased in VCD group ($p = 0.001$). Compared with the VCD group, the intensity ratio increased in VCD + E2 ($p = 0.002$) and VCD + RP1 ($p = 0.004$) but not in VCD + RP10 ($p = 0.16$) and VCD + RP100 group ($p = 0.13$) (Figure 6C).

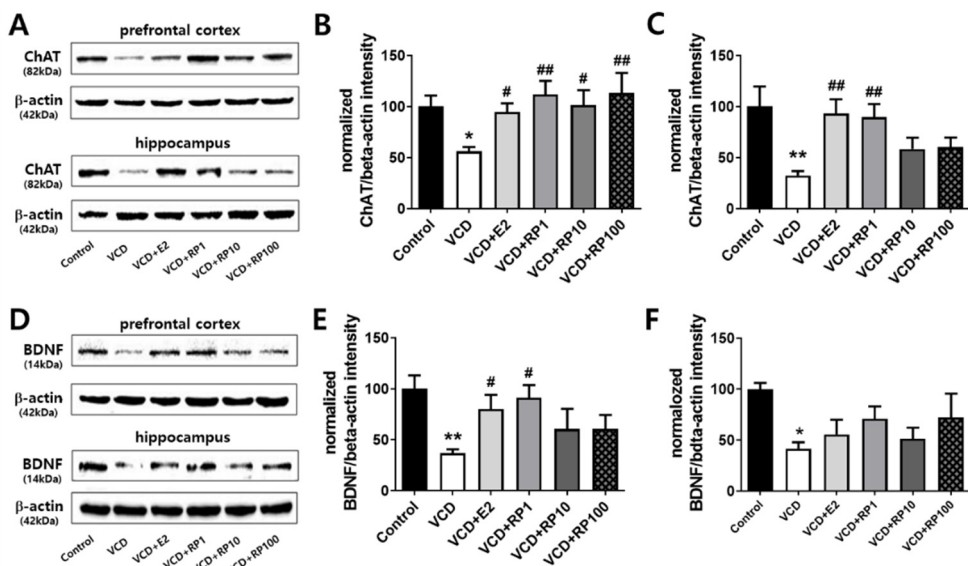

**Figure 6.** Effect of E2 and RP on the prefrontal cortex and hippocampal expression of ChAT and BDNF. (**A**) Representative blotting images of ChAT. (**B**) The ChAT band intensity ratio of the prefrontal cortex. Compared with the control group, the intensity ratio decreased in VCD. Compared with the VCD group, the intensity ratio increased in VCD + E2, VCD + RP1, VCD + RP10, and VCD + RP100. (**C**) ChAT band intensity of the hippocampus. Compared with the control group, the intensity ratio decreased in VCD. Compared with the VCD group, the intensity ratio increased in VCD + E2 and VCD + RP1 but not in VCD + RP10 and VCD + RP100. (**D**) Representative blotting images of BDNF. (**E**) BDNF band intensity ratio of the prefrontal cortex. Compared with the control group, intensity ratio decreased in VCD. Compared with the control group, intensity ratio decreased in VCD. Compared with the VCD group, the intensity ratio increased in VCD + E2 and VCD + RP1 but not in VCD + RP10 and VCD + RP100. (**F**) BDNF band intensity ratio of the hippocampus. Compared with the control group, intensity ratio decreased in VCD. Compared with the VCD group, there were no difference in intensity ratio. All data are the mean ± SEM. Control: Vehicle (sesame oil) i.p.; VCD: VCD 160 mg/kg i.p.; VCD + E2: VCD 160 mg/kg i.p + estradiol 100 µg/kg i.p.; VCD + RP1: VCD 160 mg/kg i.p + RP 1 mg/kg p.o.; VCD + RP10: VCD 160 mg/kg i.p + RP 10 mg/kg p.o.; and VCD + RP100: VCD 160 mg/kg i.p + RP 100 mg/kg p.o; * $p < 0.05$, ** $p < 0.01$ vs. Control, [#] $p < 0.05$, and [##] $p < 0.01$ vs. VCD. ANOVA, Tukey's HSD post hoc test.

BDNF expression in the prefrontal cortex showed group difference [$F(5,18) = 2.8$, $p = 0.046$]. Compared with the control group, intensity ratio decreased in VCD ($p = 0.004$). Compared with the control group, intensity ratio decreased in VCD group ($p = 0.001$). Compared with the VCD group, the intensity ratio increased in VCD + E2 ($p = 0.038$) and VCD + RP1 ($p = 0.013$) but not in VCD + RP10 ($p = 0.24$) and VCD + RP100 group ($p = 0.24$) (Figure 6E). Additionally, BDNF expression in the hippocampus showed group difference [$F(5,18) = 3.4$, $p = 0.047$]. Compared with the control group, intensity ratio decreased in VCD group ($p = 0.046$). Compared with the VCD group, there were no difference in the intensity ratio of other groups (Figure 6F). These results show that estrogen and RP can improve VCD-induced reduction in ChAT and BDNF.

*3.6. RP increased Hippocampal BAG1 Expression in VCD-Treated Mice*

Stress-regulating molecules were measured in the prefrontal cortex and hippocampus (Figure 7; $n = 6\sim8$/group). Hippocampal expression of GR showed an insignificant trend of changes in VCD-treated groups of mice (Figure 7B). BAG1 expression in the prefrontal cortex showed group difference [$F(5,36) = 3.2$, $p = 0.023$]. Compared with the control group, the intensity ratio decreased in VCD ($p = 0.04$). Compared with the VCD group, the intensity ratio increased in VCD + E2 ($p = 0.015$), VCD + RP1 ($p = 0.001$) VCD + RP10

($p$ = 0.009), and VCD + RP 100 ($p$ = 0.015) (Figure 7C). These results show that estrogen and RP can improve stress regulation through BAG1 in VCD-treated mice.

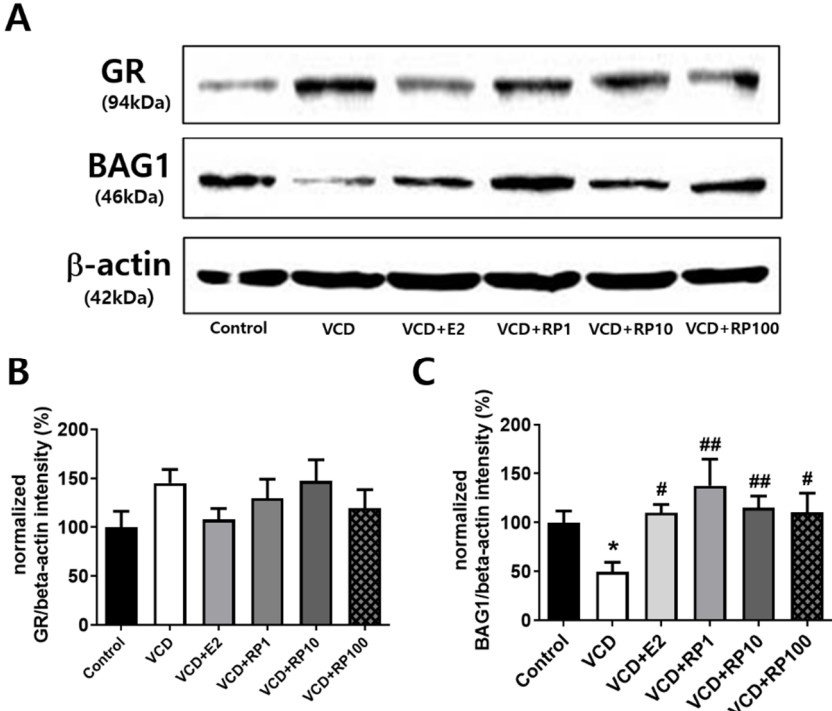

**Figure 7.** Effect of E2 and RP on hippocampal expression of GR and BAG1 in VCD-treated mice. (**A**) Representative blotting images (**B**) Band intensity ratio of GR. There was no significant difference. (**C**) Band intensity ratio of BAG1. Compared with the control group, the intensity ratio decreased in VCD. Compared with the VCD group, the intensity ratio increased in VCD + E2, VCD + RP1, VCD + RP10, and VCD + RP100. All data are the mean ± SEM. Control: Vehicle (sesame oil) i.p.; VCD: VCD 160 mg/kg i.p.; VCD + E2: VCD 160 mg/kg i.p + estradiol 100 μg/kg i.p.; VCD + RP1: VCD 160 mg/kg i.p + RP 1 mg/kg p.o.; VCD + RP10: VCD 160 mg/kg i.p + RP 10 mg/kg p.o.; and VCD + RP100: VCD 160 mg/kg i.p + RP 100 mg/kg p.o; * $p < 0.05$ vs. Control, # $p < 0.05$, and ## $p < 0.01$ vs. VCD. ANOVA, Tukey's HSD post hoc test.

## 4. Discussion

VCD-induced estradiol depletion, unlike OVX, undergoes progressive degeneration like the human menopausal process and residual tissues maintain their secretory function of other hormones [48]. Cognitive impairment, anxiety, and depression were observed in the VCD-induced estradiol depletion model [37,49]. Based on the literature, the optimal dose of VCD to accelerate estradiol depletion was 160 mg/kg for 20 days [47,48]. In this condition, the time to reach complete ovarian failure was 52 ± 2.2 days [47]. However, in this experiment, 15% of mice continued their estrous cycle after 56 days, and these mice were excluded from the experimental groups. Ovarian failure was determined by vaginal cytology when the estrus cycle no longer exhibited cycling and the diestrus stage lasted more than 10 days [43,47]. An alternative estradiol depletion model may be aged female mice. However, there is no menopause in rodents because reproductive aging in rodents does not begin in the ovaries, as it does for humans [50,51]. Therefore, aged rodents are not suitable as an estradiol depletion model [45].

Studies have demonstrated that RP may be effective in dementia and depression [37]. However, whether RP can improve dementia and depression in menopausal models needs to be tested. Tenuigenin, a component of RP, improved learning and memory in OVX mice but whether RP has antidepressant effects in the VCD-induced estradiol depletion model is unknown [52].

The serum concentration of estradiol and uterus weight decreased by VCD treatment. However, estradiol administration increased estradiol serum concentration and prevented uterine atrophy in VCD-treated mice which may be an endometrial growth-stimulating effect [53]. RP did not simulate the endometrial growth nor increase serum estradiol concentration. Phytoestrogens such as isoflavone stimulate estrogen receptors and have been suggested as a replacement for estrogen [54]. However, RP did not affect the atrophied uterus. There are two types of estrogen receptors, alpha and beta, and alpha receptors are the main receptors distributed in the mammary gland and uterus [55]. The weight increase in the uterus by estrogen in the menopause model may be an effect through the alpha receptor. While the beta receptor effect of RP cannot be confirmed, at least the effect on the alpha receptor seems to be insignificant. Although estrogen receptor-modulating activity was shown in many medicinal plants, such effect was not clearly disclosed in RP [56]. Therefore, RP appears to have no or weak phytoestrogenic effect.

The estrogen deficiency in menopause is one of the factors that cause cognitive decline [57], and estrogen enhances ChAT activity [58]. By chronic VCD injection, working memory, spatial memory, and ChAT expression in the prefrontal cortex and hippocampus were decreased. Estrogen and RP administration improved the cognitive decline and ChAT expression in VCD-treated mice. RP 1 mg/kg and 100 mg/kg improved working memory in the Y-maze test. In the MWM test, swimming time in the target zone was improved by estrogen and all doses of RP treatment. Moreover, the ChAT expression was improved in the prefrontal cortex by all given doses and in the hippocampus by 1 mg/kg RP. Decreased ChAT after VCD treatment may explain memory impairment in these mice indicating a reduced amount of acetylcholine. Previously, RP was reported with improving effects on learning and memory and ChAT activity [59]. As estrogen and RP restore the VCD-induced reduction in ChAT expression, acetylcholine production may have increased.

Menopausal estrogen deficiency is also related to depression [21]. Estrogen increases BDNF expression, which contributes to emotional stability [20,23]. By chronic VCD injection, depression-like behaviors and BDNF expression in the prefrontal cortex and hippocampus were decreased. A supplement of estrogen and RP administration improved the depressive behaviors and BDNF expression in VCD-treated estradiol depletion model mice. Among the components of RP, those known to have antidepressant effects include polygalasaponins, tenuifolin, tenuigenin (senegenin), and DISS [60]. In our RP extract, there was a significant amount of DISS, but tenuifolin was hardly detected. This is because tenuifolin and tenuigenin form the backbone structure of polygalasaponins, and they are detected in their glycosylated forms [61]. It is known that these sugar structures are removed by intestinal bacteria and reduced to tenuifolin and tenuigenen for absorption. Therefore, it seems that the antidepressant effect of RP extract was shown by tenuifolin, tenuigenin, or DISS.

Additionally, BDNF expression improved in the prefrontal cortex by 1 mg/kg of RP. Previously, RP demonstrated antidepressant-like effects [34] and increased the expression of BDNF [20,35]. However, RP showed rapid onset antidepressive effect which was most potent at 0.1 mg/kg [35]. This may explain why RP increased BDNF expression by 1 mg/kg but not by higher doses. On the other hand, there was a subtle difference in the FST and TST results, that all doses of RP effectively exerted antidepressive-like effects, but in the FST, 1 mg/kg was not effective. This was different from previous findings and could be specific to VCD-induced conditions. BDNF has an important role in antidepressant effects [62]. However, in this study, BDNF expression was not consistent with the degree of antidepressant action, so it seems that the antidepressant action of RP was not related to BDNF. Instead, RP is likely related to GluR1 and mTOR signaling [35]. This is a characteristic that appears in drugs with rapid-onset antidepressant action such as ketamine, and RP is also expected to have rapid-onset properties [35]. As ketamine exhibits antidepressant action only at low doses, this may partially explain why the dose–response pattern of RP is ambiguous.

The hippocampus regulates stress response through the modulation of the HPA axis [27]. Stress hormone binding to the GR downregulates BDNF and induces anxiety-like behaviors [31,63]. BAG1 suppresses GR-induced gene transcription and as demonstrated in the literature, RP increased the expression of BAG1 in the hippocampus [2]. BAG1 overexpression under an NSE-promoter promoted recovery from stress-induced behavioral disorders [33]. By chronic VCD injection, BAG1 expression in the hippocampus was decreased. A supplement of estrogen and RP improved BAG1 expression in VCD-treated mice. In contrast, GR expression did not change with statistical significance. Because posttranscriptional regulation contributes significantly to GR activity, various emotional changes may occur without altered GR expression [64]. Reduced BAG1 expression may have caused GR-hyperactivity in VCD-treated mice, which then normalized after estrogen and RP administration. By contrast, estrogen also improved BAG1 expression. This finding may explain the loss of stress resistance after menopause [65]. As VCD ovotoxicity is associated with the bcl-2/BAX pathway, increased expression of BAG1 may have reduced VCD toxicity [66]. However, in the experiment, RP was administered after 5 weeks of the termination of VCD treatment. Therefore, if BAG1 was induced only after RP treatment, the increase in BAG1 did not affect the estrogen deficiency-inducing process of VCD.

There are some limitations to our findings. First, because female mice do not experience menopause, the mouse menopause model is difficult to apply to humans. Second, if the concentration of neurotransmitters such as acetylcholine or the activity of acetylcholinesterase were measured, it would have been possible to draw a clearer conclusion about the effect of RP. The significance of this study was that the dementia preventive and antidepressive-like effect of RP was tested in an estrogen deficiency model. There are many causes of dementia and depression, of which estrogen deficiency can also be a cause. Early estrogen supplementation therapy can effectively prevent these conditions, however, as there are side effects, reducing the dose of estrogen and using other supplements may be helpful. Effective for the treatment of dementia and depression in estrogen deficiency, RP serves this purpose well.

## 5. Conclusions

Estrogen deficiency by VCD decreases ChAT, BDNF, and BAG1 and consequently leads to cognitive decline and depression-related behaviors. Estrogen supplement and RP administration can increase ChAT, BDNF, and BAG1 deficits in the prefrontal cortex or the hippocampus and improve cognitive and depressive behaviors in VCD-treated mice. In particular, RP increased the expression of BDNF in the prefrontal cortex, and BAG1 in the hippocampus. These results implicate the possible benefit of using RP as a supplement of estrogen to improve cognitive and depressive symptoms in postmenopausal women.

**Author Contributions:** Conceptualization, H.H. and S.M.; methodology, J.C.; validation, H.K.K., M.A.K.; formal analysis, B.I.K.; investigation, G.H.; data curation, H.K.K., M.-J.J.; writing—original draft preparation, S.-Y.C.; writing—review and editing, S.M.; visualization, J.C.; supervision, H.H. All authors have read and agreed to the published version of the manuscript.

**Funding:** This work was supported by the BK21 plus program "AgeTech-Service Convergence Major" through the National Research Foundation (NRF) funded by the Ministry of Education of Korea [5120200313836].

**Institutional Review Board Statement:** The experimental procedures were performed in accordance with the animal care guidelines of Kyung Hee University in Institutional Animal Care and Use Committee (KHUASP(SE)-19-038).

**Informed Consent Statement:** Not applicable.

**Data Availability Statement:** Additional data and information can be obtained by the corresponding author on request.

**Conflicts of Interest:** The authors declare no conflict of interest.

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
