# Peer review of "Effects of Radix Polygalae on Cognitive Decline and Depression in Estradiol Depletion Mouse Model of Menopause"

_cimb, doi:10.3390/cimb43030118_

Round 1

Reviewer 1 Report

Radix Plogalae (PR) improved memory and depressive behaviors in a model of menopause. Although the results are interesting, I have following questions and suggestions.

  1. In Introduction, you mention that "an improvement in the method to prevent cognitive decline in menopause is needed". In this study, however, PR was administered 5 weeks after VCD injection. Was memory of the mice impaired at this time? Did PR prevent or recover from memory impairment? 
  2. Ref 37 showed that VCD injection increased LH and FSH, and decreased androstedione. Are these phenomena seen in human menopause? Did these phenomena influence on memory and depressive behaviors?
  3. Please show results of HPLC analysis for PR.
  4. In Figure 2B, what are differences among 1-4? Is everything estrus? How did you determine estrus in VCD-injected mice?
  5. Is it OK that BDNF and BAG1 do not relate to memory recovery and attenuation of depressive behaviors by PR? If so, please mention so in conclusion. Please discuss more clearly how PR improved memory and depressive behaviors in VCD-treated mice. 

Author Response

1.Thank you for the useful and valuable comments.

Because menopause is a gradual process, ovariectomy, in which estrogen drops sharply, is not appropriate to model menopause. On the other hand, the VCD model, in which the concentration of estrogen is gradually reduced, better reflects the state during menopause, so it was used in this experiment. Induction of menopause was checked by observing the cyclic changes of endometrial cells acquired from vaginal smear, as shown in Figure 2.

After 3 weeks of daily VCD injection, 5 weeks of non-treated incubation period, and 3 weeks of RP feeding, working memory was decreased by VCD group. Estradiol and RP 1 and 100 mg/kg improved the VCD-induced deficit in working memory (Figure 4A). Also, in the Morris water maze, VCD decreased the performance of spatial memory, and estradiol and RP restored the deficit of spatial memory (Figure 4B).

2. I was not aware of the findings of VCD on LH, FSH and androstenedione. The serum level of FSH and LH is known to increases after menopause due to the lack of estrogen’s feedback inhibition on gonadotrophin production. Also, due to the reduced production of androstenedione from the ovary, circulating androstenedione concentration decreases, but its influence on the body becomes larger due to the lack of estrogen. While increase in circulating FSH had no effect on cognitive and depressive behaviors (Journal of Women’s Health 2007 16(3) 331-344) quite many papers were reporting that high circulating levels of LH and androstenedione can impair cognitive and depressive behaviors. This may be an interesting aspect to address in our study, as if RP can also correct these changes and apply for antidepressant remedies in menopausal women.

3. Sorry for the inconvenience. We discovered the wrong paper was cited for the HPLC analysis data in our previous publication. This is the HPLC analysis (https://doi.org/10.5607/en.2018.27.3.200)

image separately provided (HPLC.pdf)

4. In Figure 2A, the morphological changes of endometrial cells during the 4 estrus cycle in normal mice is presented. In Figure 2B, there were no specific cytology pattern indicating the estrus phase (2A-2), in which the cells grow larger. After VCD treatment, there were only cell patterns showing the metestrus and diestrus stages.

5. We think that there was a memory recovery and attenuation of depressive behaviors according to the behavioral measurements, but in biochemical analysis, changes in BDNF or BAG1 were not as expected. While it is true that memory improved when BDNF expression increases, but all the improvement of memory cannot be a result of BDNF increase. Actually, BDNF was increased in the prefrontal cortex by RP 1mg/kg, which dose showed improvement in working memory and spatial memory. Also increase in BAG expression may have reduced the effect of stress hormones at least in the hippocampus. We added some clarity on this point in the conclusion section.

Reviewer 2 Report

 The current manuscript showed that Radix Polygalae (RP) improves depression and cognitive function using a mouse model of menopause induced by VCD. Although the mechanistic investigation is not sufficiently performed, it seems that the content is suitable for Current Issues in Molecular Biology. I request some revise below before publication.

 Regarding the discussion on the mechanism of RP, the authors could strengthen the following points.

I) Based on the changes in uterine weight, the authors conclude that RP has no estrogen-like effects. However, changes in uterine weight are mainly influenced by estrogen α receptors, while many kinds of phytoestrogens activate β receptors. Therefore, as a mechanism of action of RP, the estrogen-like effect of RP needs to be further investigated.

II) It would be better to discuss the relationship between the amount of tenuigenin in the RP used in this study and the previously reported amount of tenuigenin when it shows antidepressant effects.

Author Response

1.

We totally agree on your strengthening points and reinforced the contents in the manuscript.

As you stated, ERalpha is mainly distributed in the mammary gland, uterus, ovary (thecal cells), bone, and liver, while ERbeta is found mainly in the bladder, ovary (granulosa cells), colon, and immune system, if RP does not bind to ERalpha, there will be no uterine effect. This means we cannot excluded any ERbeta effect of RP. But as RP is not reported to have potent estrogen receptor modulating activity (Korean Journal of Pharmacognosy 2006 37(1) 21-27), although many papers report cognition improving function (Evidence-based complementary and alternative medicine 2012 https://doi.org/10.1155/2012/692621), RP may improve cognition independent to estrogenic effects.  

2.

Tenuigenin (senegenin) at the dose of 4 and 8 mg/kg showed antidepressant-like effects in the CUMS model (Int immunopharmacol 2017 53. 24-32), and a cognitive improvement effect (reference 52) in the ovariectomy model at 4 mg/kg. In our report, teniofolin was previously measured by HPLC, and only a very low concentration was detected. This was because tenuifolin exists as glycosylated forms called polygalasaponins which doesn’t form a single peak in the HPLC. But, DISS was detected in significant amounts in the extract. DISS showed antidepressant-like effect at 5-20 mg/kg in chronic mild stress exposed rats (ref 41). Therefore, DISS may have mainly contributed to the antidepressive effects. However, tenuifolin and tenuigenin reduced from polygalasaponine by enteric bacteriae may also have been involved in the antidepressant action.

Reviewer 3 Report

I read the manuscript with much interest. I found it technically really good with appropriate laboratory analyses and statistical approach. Importantly, it has a nice practical soundness, however clinical trials are needed.

Presented data basicly supports the conclusions. Experiments have been conducted properly, with appropriate controls, however there is one issue that I strongly encourage authors to address before the paper is ready to be published:

Please report the study according to SYRCLE protocol

 I recommend to add mRNA expression on tested factors

Author Response

1.

Thank you for your valuable advise to improve the quality of this manuscript. I understand that SYRCLE protocol applies to systematic review of animal studies. But this manuscript reports the finding of a single animal study. If I have any misunderstanding, please let us know what we can do.

2.

It will be very informative and useful by providing mRNA expression of ChAT, BDNF, GR, and BAG1 to support our conclusions. But we think the protein expression results are more definitive to evaluate the molecular changes in this model system. It would be best to provide both data, but if only one is to be selected, we think it is better to present protein than mRNA measurements. Second is the bigger issue. Unfortunately, we only have 7 days to submit the revised manuscript.

Round 2

Reviewer 1 Report

No comment.